



# Inferring the role of IPO phase dependencies and extratropical internal variability on the tropics

Mark A. Collier[1], Dylan Harries[2], and Terence J. O'Kane[3]

[1]CSIRO Oceans and Atmosphere, Aspendale, Melbourne, Victoria, Australia
[2]South Australian Health & Medical Research Institute, Adelaide, Australia
[3]CSIRO Oceans and Atmosphere, Battery Point, Hobart, Tasmania, Australia

**Correspondence:** Mark A. Collier (mark.collier@csiro.au)

**Abstract.** Regime dependencies and Granger causal relationships between tropical and extratropical teleconnections are inferred using Bayesian structure learning. Using ERA5 data, an examination of the differences between the learned graphical structures during particular phases of the Interdecadal Pacific Oscillation (IPO) are used to infer the role of the background state on interactions between the major climate teleconnections. These relationships present a clear regime dependency on the

phase of IPO. In the positive phase, IPO autocorrelations are weak whereas Indian Ocean Dipole (IOD) and El Niño Southern Oscillation (ENSO) autocorrelations and the influence of the Madden Julian Oscillation (MJO) are indicative of an enhanced Walker circulation. In contrast, during the negative phase, IPO autocorrelations are strongest with evidence of an enhanced role for extratropical teleconnections on the tropics. Exclusion of MJO removes important tropical-extratropical influences while increasing posterior edge weights between ENSO, the IPO and IOD. Our analysis reveals the dependence of the ENSO auto-

correlation on the phase of the background IPO state, and the role of the MJO as being key to link the extratropical tropospheric modes (PNA, NAO) and equatorial surface ocean temperatures (IOD, ENSO) and as a consequence convection.

## 1 Introduction

The complexity of the climate system is most evident in the interactions between the respective internal modes of variability, otherwise known as teleconnections, modes that are distinct in both geographical location and timescale. Whereas internal

variability occurs on timescales from synoptic to inter-annual, the longer time- and larger spatial-scale multi-decadal modes act as background states, most notable of which are the Inter-decadal Pacific Oscillation (IPO) and Atlantic Multi-decadal Oscillation (AMO). The extent to which these low frequency background states impose regime dependencies on the shorter timescale internal modes, and therefore their intrinsic predictability, remains largely an open question. Quantifying (Granger) causal relationships from reanalyses or data from dynamic general circulation models (GCMs) is challenging due the dimensionality and

large sample sizes required to determine accurate statistical measures from often non-stationary timeseries. Recent work by the authors (Harries and O'Kane, 2021; O'Kane et al., 2024) has shown that the current generation of GCMs often exhibit large systematic biases in representing autocorrelations and edge weights in graphical representations of their associated structural causal models. The ability to quantify model biases and their uncertainties in relation to observational products from a causal



perspective is key to determining not only the source of given model error(s) but to enhancing our understanding of the climate
system thereby aiding the development of better predictive systems.

In this study we seek to quantify differences in the inferred dynamic Bayesian network structures describing interactions between the major modes of tropical ocean and extratropical atmospheric variability for alternative phases of the dominant background state of the Pacific Ocean over multidecadal timescales. To this end, we examine models including an index of the Interdecadal Pacific Oscillation (IPO) during representative phases of the IPO. The IPO (Power and Coleman, 2006; Meehl
et al., 2010; Dong and Dai, 2015; Heidemann et al., 2023) characterizes low frequency sea surface temperature (SST) variability on decadal timescales over the entire Pacific basin, incorporating variations due to the El Niño Southern Oscillation (ENSO), the North Pacific Decadal oscillation (PDO) (Mantua et al., 1997), and the South Pacific Decadal Oscillation (SPDO) (Chen and Wallace, 2015). The IPO, as characterized by the Tripolar Index (TPI) of Henley et al. (2015), exhibits a distinct 'tripole' pattern of SST spanning the Pacific Ocean from the Northern Hemisphere subtropics to the Southern Hemisphere subtropics.
Over the decades covered by reanalysis data sets, debate on the mechanisms driving substantive variations in the phase of the IPO has been focused on the relative role of external forcing in the form of anthropogenic warming (Kosaka and Xie, 2013), versus internal variability (Risbey et al., 2014; Liguori et al., 2020; Sun et al., 2021).

The phase of the IPO has been shown to have impacts both globally and at a regional scale. For example, periods of accelerated global warming during the last decade of the twentieth century or the recent warming hiatus earlier this century
have been shown to align with positive and negative phases of the IPO (Meehl et al., 2013; England et al., 2013). Australian rainfall is related to IPO phase which can clearly have a bearing on flood threat as well as agricultural outputs and heightened levels of social anxiety. During the boreal winter (DJF), the IPO in concert with low frequency SST variations in the Atlantic i.e., the Atlantic Multidecadal Oscillation, are known to induce synoptic Rossby wave patterns over Eurasia, which act to modulate the East Asian trough and jet stream, and blocking highs over the Urals and Siberia (Yang et al., 2024). Recent
studies point to the role of the IPO on surface winds over the densely populated South Asian region (Yuan and Shen, 2025). In addition the role of the IPO in modulating the Pacific Walker (Bjerknes, 1969) and Hadley (Webster, 2004) circulations has started to be recognized; for example, Wu (2024) point to internal variations due to IPO phase change as the likely determinant for the recent strengthening of the Pacific Walker circulation and widening of the Hadley Cell. That said, the role of the IPO in modulating variations of the respective internal tropical modes remains open but one of increasingly recognized importance. In
this study we apply, for the first time, the methodology of Bayesian structure learning to infer conditionally (Granger) causal directed acyclic graphical network representations of the most probable lagged relationships between the tropics and the major modes of variability of both hemispheres and their dependence on IPO phase. The introduction of the IPO into the inference set of random variables allows investigation of the role that the extratropics play in determining tropical variability. We pay particular attention to changes in autocorrelation and the inferred respective roles of the teleconnections in communicating
variability between tropics and extra-tropics.

The use of structure learning has been only recently been proposed as a means of evaluating the ability of coupled general circulation models (CGCMs) to reliably simulate the relationships between major modes of climate variability (Vázquez-Patiño et al., 2020; Nowack et al., 2020; O'Kane et al., 2024). Constraint-based (Spirtes and Glymour, 1991) or score-based (Geiger



and Heckerman, 1994) methods may be used to learn the time lagged relationships and autocorrelations for the selected modes,
which may be represented graphically as a dynamic Bayesian network (DBN). The inferred Granger causal relationships
provide an intuitive simplified representation of the underlying dynamics (note that, in the following we always use "causal" in
the sense of Granger). Where the structure of the network is not known a priori, it must be inferred using appropriate criteria,
e.g., the correlation or mutual information (Tsonis et al., 2007; Donges et al., 2009) between time series associated with each
given node, thresholded according to the required level of statistical interdependence between node pairs. For the Bayesian
directed network models presented here, the graph encodes the set of independence relationships between the timeseries of
climate teleconnections, by identifying a corresponding set of random variables with nodes in the network and encoding the
joint probability density function (PDF) in the included edges.

In the absence of expert knowledge, constraint-based algorithms (Colombo and Maathuis, 2014), in which the set of
edges is determined starting from a series of conditional independence tests, have proved popular (Ebert-Uphoff and Deng,
2012a, b, 2017; Runge et al., 2019). Such methods can flexibly incorporate linear or non-linear conditional independence tests
together with predefined constraints, allowing for non-linear dependence structures to be estimated from data (Hlinka et al.,
2013). However, the inclusion of edges on the basis of an (initially arbitrary) significance level together with multiple testing
adjustments makes assessing the level of confidence in the inferred networks difficult. Instead, various strategies, such as vary-
ing the hyperparameters of the structure learning algorithm (Ebert-Uphoff and Deng, 2012a) or the form of independence test
used (Hlinka et al., 2013), are often used to identify robust links in a fitted network. Structure learning methods have recently
been applied as a model evaluation tool by Nowack et al. (2020), in which a constraint-based approach was used to compare
the graph structures obtained in Coupled Model Intercomparison Project version 5 (CMIP5) models (Taylor et al., 2012) to
those found from reanalyses.

In previous applications of Bayesian structure learning to reanalysis (Harries and O'Kane, 2021) and model (O'Kane et al.,
2024) data, the DBN structure has been assumed to be constant in time (i.e., time-homogeneous). In practice, in a multiscale
climate system, the various modes of variability are known to interact across spatiotemporal scales, which, in combination
with non-stationary forcing (e.g., anthropogenic warming), has the potential to manifest as changes in the network structure or
the corresponding vector autoregressive (VAR) model parameters (Wu et al., 2018; Saggioro et al., 2020). In the current work,
we approach the computationally challenging task of direct inference of time-varying networks via a score-based approach
by examining time-homogeneous networks fitted to time periods corresponding to the distinctly different but clearly defined
IPO background states. Due to constraints on the observational record, there occurs only one positive IPO (+IPO: 1977-1999)
and one negative IPO (-IPO: 1948-1976) phase of sufficient length to allow for accurate fitting of a Bayesian structural causal
model. In order to sample from a wider range of possible background states, we therefore additionally examine an ensemble of
historical CMIP6 model simulations from the ACCESSCM2 model (Bi et al., 2020). By approaching structure learning based
on Bayesian inference we allow for explicit quantification of uncertainty (Cooper and Herskovits, 1992; Heckerman et al.,
1995; Harries and O'Kane, 2021; O'Kane et al., 2024; Zhou et al., 2024), which may potentially be considerable given the
restricted instrumental record. For a comprehensive discussion of constraint based versus Bayesian structure learning methods
see Harries and O'Kane (2021).



In Section 2 we describe the data and derivation of the climate indices. In Section 3 we describe the Bayesian structural causal
models (SCMs) methods and in Section 4 discuss the resulting DBNs. Lastly our conclusions are summarized in Section 5.

## 2   Data and Diagnostics

The analyzed teleconnections are calculated from the fifth generation ECMWF reanalysis (ERA5) This reanalysis was chosen
because it provides an up-to-date, complete set of variables at sufficient temporal resolution from which to calculate best esti-
mate indices of the major modes of climate variability over the decades post-1940. Hourly ERA5 data with a spatial resolution
of $0.25°$ longitude $\times 0.25°$ latitude ($1440 \times 721$ points, respectively) are averaged to daily and, by selecting every fourth point
in each horizontal direction, reduced to $1°$ longitude $\times 1°$ latitude ($360 \times 180$) resolution. In addition, low-frequency variability
is very well represented for most of the atmosphere, with temperature patterns that agree well with those of the other major
reanalysis systems, including JRA-55 (Kobayashi et al., 2015) and MERRA-2 (Gelaro et al., 2017).

We further examine three historical simulations spanning the years 1850-2014 generated by the ACCESSCM2 GCM (Bi
et al., 2020) and submitted to the CMIP6 (Eyrng et al., 2016). The ACCESSCM2 horizontal Gaussian atmospheric variables
are linearly interpolated, and the tri-polar ocean gridded data bi-linearly interpolated, to the common $1° \times 1°$ degree resolution
of the ERA5 reanalysis. To fully examine tropical interactions we need to understand the Madden Julian Oscillation (MJO),
which is considered an eastward moving 'pulse' of cloud and rain occurring every 1 to 2 months. The MJO can be accurately
defined by the $RMM1/RMM2$ indices of Wheeler and Hendeon (2004). Unlike for ERA5, the $RMM1/RMM2$ were
not able to be calculated for the ACCESS model as the required daily variables are unavailable and so are not present in
the inferred ACCESSCM2 networks. For this reason two further Bayesian network experiments using ERA5 inputs with the
$RMM1/RMM2$ indices removed were conducted to enable an appropriate comparison with the ACCESS model experiments.

For the computation of the Pacific South American ($PSA$) pattern (Table 1), the loading pattern was computed from the
monthly anomalies of 500hPa geopotential height ($Z_g^{500hPa}$) relative to a 1979-2001 base period. Our investigations using this
approach on ERA5 for which there were both daily and monthly inputs indicated the impacts of reduced temporal resolution
were minor. In addition, the Northern Hemisphere tropospheric teleconnections ($NHTELE$) calculated via k-means, were
also computed using monthly anomalies of $Z_g^{500hPa}$ rather than the traditional daily values, again with apparent minor impacts
on the patterns of the cluster centroids and in comparison to reference indices. The considered climate teleconnection indices
and their dependencies are described in Table 1.

Although a principal component (PC) based approach to computing the IPO produces a robust signal of the SST variability
associated with the dominant Pacific basin wide variability, and thereby potentially mitigates systematic biases present in some
models, the fixed latitude/longitude rectangular boxes of the IPOTPI has been found by us to adequately capture the signal in
both our reanalysis and models under investigation, with numerous computational advantages as described by Henley et al.
(2015). It is defined by:

$$IPOTPI = SSTA_2 - \frac{(SSTA_1 + SSTA_3)}{2}, \tag{1}$$





**Table 1.** Input variables for calculation of climate indices

| Tropical | Northern Hemisphere | Southern Hemisphere |
|---|---|---|
| MEI: multivariate ENSO index | AO: Arctic Oscillation | SAM: Southern Annular Mode |
| ENSO: El Niño Southern Oscilation | NAO(+,-): North Atlantic Oscillation | PSA(1, 2): Pacific-South American pattern |
| IOD: Indian Ocean Dipole | PNA: Pacific-North American pattern | |
| RMM(1 and 2): Real-time Multivariate MJO index | AR: Atlantic ridge | |
| IPO: interdecadal Pacific Oscillation | SCAND: Scandinavian blocking | |

| Raw Variable | Abbreviation |
|---|---|
| 500hPa geopotential height | $Z_g^{500hPa}$ |
| 850 and 200hPa zonal winds | $U^{850hPa}, U^{200hPa}$ |
| Mean sea level pressure | $MSLP$ |
| Top-of-atmosphere outgoing longwave radiation | $OLR$ |
| 10m zonal and meridional wind | $U^{10m}, V^{10m}$ |
| Sea-surface temperature | $SST$ |

| Index | Variable(s) |
|---|---|
| $AO, SAM, PNA, PSA,$ $NHTELE1/AR,$ $NHTELE2/NAO+, NHTELE3/NAO-,$ $NHTELE4/SCAND$ | $Z_g^{500hPa}$ |
| $RMM1^*, RMM2^*$ | $U^{850}, U^{200}, OLR$ |
| $IPO, IOD$ | $SST$ |
| $MEI$ | $U^{10m}, V^{10m}, MSLP, SST, OLR$ |



**Table 2.** Datasets and representative IPO phase periods

| Dataset | -IPO years | +IPO years | All years |
|---|---|---|---|
| ERSST | 1948–1976 | 1977–1998 | 1940–2025 |
| ERA5 | 1948–1976 | 1977–1998 | 1948–1998 |
| ACCESSCM2R1 | 1915–1944 | 1945–1974 | 1915–1974 |
| ACCESSCM2R5 | 1960–1989 | 1930–1959 | 1930–1989 |
| ACCESSCM2R7 | 1867–1896 | 1915–1944 | N/A |

where SSTA is the monthly SST anomaly (relative to same base or entire period) and 1, 2 and 3 denote different regions which are latitude/longitude horizontal boxes defined by the edges $25°N \rightarrow 45°N/140°E \rightarrow 145°W$, $10°S \rightarrow 10°N/170°E \rightarrow 90°W$ and $50°S \rightarrow 15°S/150°E \rightarrow 160°W$, respectively.

To examine causal relationships in physically well established indices of the tropical-extratropical region shown in Table 1 and as used in our previous studies (Harries and O'Kane, 2021; O'Kane et al., 2024), augmented here with the addition of IPOTPI following Eq. 1, we extracted relevant datasets from the ERA5 reanalysis and ACCESSCM2 model simulations. Hereafter we refer to the IPOTPI as simply the IPO. Periods for positive and negative phases of the observed IPO are shown in Figure 1a and provided in Table 2. The monthly ERA5 IPO is found to be very similar to that derived from the extended reconstruction SST (ERSST) analysis version 5 (Huang et al., 2017), as expected. The 30 year moving window results for ERA5 & ERSST are almost identical for the positive IPO phase, but are somewhat different during the negative phase due to less consistency in the observed record in the first two-thirds of the twentieth century. Low pass filtered results, obtained by retaining lowest order harmonics (7 modes), are added to this figure, where the moving window approach matches well in the positive phase but with some discrepancy during the negative IPO phase, although both are still clearly of a negative character. Limitations of the low pass method are exemplified beyond 2010 where the observed IPO phase is clearly trending in a negative direction, however, this is generally in agreement with the straight-forward moving window approach that we have implemented here to identify clear signed periods of IPO phase.

The observed IPO phases over the second half of the twentieth century are provided in Table 2 and for this we have taken the phase switch year to be between 1976 and 1977. To isolate IPO regime shifts as simulated by the ACCESSCM2 model, we have examined the time-series over the full historical experiment, 1850 to 2014. To identify significant (when compared to what is observed) IPO phases we compute a 30 year moving window on the IPO and manually identify 30 year periods for each where they exist. These are provided in Table 2 for ACCESS and ERA5, and diagramatically in figure 1a for the ERA5. The chosen experiments with well defined IPO phases are numbered 1, 5 and 7. The DBN experiments are run separately for each IPO phase as well as for all years together, with the exception as follows. For the reanalysis and the ACCESSCM2{1,5} realizations the IPO phases are temporally contiguous, however, with the ACCESSCM2R7 realization the IPO phases are separated by a







**Figure 1.** (a) Monthly, 30 & 7 year moving window low pass filtered IPO index for ERSST and ERA5 spanning years 1940-2024. Significant negative and positive phases of the IPO occur over the periods 1948-1976 and 1977-1998 respectively. The unfiltered ERSST monthly IPOTPI is pre-computed from Henley et al. (2015). The Niño3.4 index has been calculated from the monthly anomalous ERA5 SST data. Posterior probabilities for edges from ERA5 data over the period 1948-1999: (b) excluding IPO; (c) including IPO & (d) excluding MJO.





neutral phase lasting a number of decades. This set of three will allow us to comment on regime shifts versus independent regime events. To better observe the slowly varying component of the IPO in the monthly data often a low pass filter or similar is applied, however, in this case we have left the monthly data unfiltered to be compatible with the other indices, which exhibit significant spectral energy at a range of higher frequencies. The IPO and ENSO indices, for example Niño3.4, co-vary strongly, however, there are exceptions especially during negative IPO events when the extratropical SST anomalies dictate the overall

magnitude and sign of the IPO index. During positive IPO events SSTs in the tropics dominate the index, where negative and positive extremes of the IPO index are generally well represented by the Niño3.4 index.

## 3   Methods

Causal relationships between the selected teleconnections are inferred under the assumption that the system may be adequately described by linear VAR models, represented graphically as DBNs (while the inclusion of a moving average component may

yield a better fit for some modes, the use of VAR models somewhat simplifies model implementation). Here we briefly summarize the method used to fit the models, shown schematically in Figure 2. Further details may be found in O'Kane et al. (2024).

The observed value of each index $i$ in month $t$, $Y_t^i$, is assumed to be normally distributed with mean $\mu_t^i$ and variance $\tilde{\tau}_i^{-2}$ (panel 1 of Figure 2), with

$$\mu_t^i = \beta_0^i + \sum_{j=1}^{p_i} \beta_{(k_j, \tau_j)}^i Y_{t-\tau_j}^{k_j}. \tag{2}$$

The collection of indices $\mathrm{pa}_G(Y_t^i) = \{Y_{t-\tau_j}^{k_j}\}$ upon which $Y_t^i$ depends is referred to as the parent set of $Y_t^i$. Graphically, the model may be represented as a network in which the nodes are the random variables corresponding to the indices and their lagged values, up to some maximum lag $\tau_{\max}$. Directed edges are drawn from the node for each member of the parent set to the node representing the index at time $t$ (the child node). Contemporaneous edges are assumed to be absent, ensuring acyclicity of

the graph at the cost of excluding possible interaction taking place on time-scales shorter than the monthly sampling frequency. The resulting graph is assumed to be constant over time, and thus corresponds to a time-homogeneous DBN. To investigate possible changes in the structural relationships associated with changes in the background state, separate models were fitted to periods of positive and negative IPO phase, in addition to the full timeseries. In this respect, we assume that the prescribed IPO phases represents period in which the relationships between the different modes are stationary, and can be inferred by stratifying

on IPO phase. While doing so invariably results in some loss of power due to reduced sample size, and may introduce artefacts arising from discretization of what is likely a continuous underlying process, the use of separate models is a simple approach to allow an initial investigation into variability in teleconnections with respect to the background state without dramatically increasing model complexity. The implementation of more realistic non-stationary models may be facilitated by the use of alternative model specifications (Zhou et al., 2024), and is left for future study.

Both the parent sets (i.e., the graph structure) and associated VAR parameters are taken to be unknown, and are inferred from the historical data or model runs. Independent, conjugate normal-gamma priors are assumed for the VAR parameters





## Structure Learning, Conditional densities and prior distributions

### 1: Define generative model and priors

Conditional on the parents $pa_G(Y_t^i)$
each index $Y_t^i$ is normally distributed with
mean $\mu_t^i$ given by a linear function of the parent variable values

$$Y_t^i | \mathrm{pa}_G(Y_t^i), \tilde{\tau}_i^2 \sim N(\mu_t^i, \tilde{\tau}_i^{-2})$$

$$\mu_t^i = \beta_0^i + \sum_{j=1}^{p_i} \beta_{(k_j,\tau_j)}^i Y_{t-\tau_j}^{k_j}$$

Conjugate normal-gamma priors specified for the regression coefficients
$\beta_{(k_j,\tau_j)}^i$ and the conditional precision $\tilde{\tau}_i^2$

$$\tilde{\tau}_i^2 \sim \mathrm{Gamma}(a_{\tau_i}, b_{\tau_i})$$

$$\beta_0^i | \tilde{\tau}_i^2 \sim N\left(0, \frac{\nu_i^2}{\tilde{\tau}_i^2}\right)$$

$$\beta_{(k_j,\tau_j)}^i | \tilde{\tau}_i^2, \mathrm{pa}_G(Y_t^i) \sim N\left(0, \frac{\nu_i^2}{\tilde{\tau}_i^2}\right) \quad j = 1, \ldots, p_i$$

### 2: Learn graph structure and model parameters

Given data $D = y_1, \ldots, y_T$ where $y_t$ denotes the values of the random variables
$Y_t = (Y_t^1, \ldots, Y_t^n)^T$ at time t, learning the structure $G$ and parameters $\theta$, where parameters $\theta$
consist of the coefficients $\beta_{(k_j,\tau_j)}^i$, and the conditional precision $\tilde{\tau}_i^2$

$$P(\theta, G | D) = P(\theta | G, D) P(G | D)$$

### 3: Score based approach to model selection

Sampling from the full posterior distribution of possible graphs $P(G|D)$ requires
evaluation of the marginal likelihood

$$P(D | G) = \int d\theta P(D | G, \theta) P(\theta | G)$$

### 4: Apply MCMC

Given a current candidate structure $G$, the sampling scheme proceeds by proposing a new
structure $G'$ according to a proposal distribution $q_G(G'; G)$.
The proposal $G'$ is accepted with probability

$$\alpha = \min\left\{1, \frac{q_G(G; G')}{q_G(G'; G)} \frac{P(D|G')}{P(D|G)} \frac{P(G')}{P(G)}\right\}$$

Set maximum time-lag $\tau_{max}$ and impose a maximum size $p_{max}$
on the allowed parent sets in order to sparsify the networks

$$P(\mathrm{pa}_G(Y_t^i)) = \begin{cases} \left[\sum_{j=0}^{p_{\max}} \binom{n\tau_{\max}}{j}\right]^{-1}, & |\mathrm{pa}_G(Y_t^i)| \leq p_{\max}, \\ 0, & \text{otherwise.} \end{cases}$$

We also adopt a uniform proposal density on graphs $G'$
in the neighbourhood of the current graph $G$,

$$q_G(G', G) = \begin{cases} \frac{1}{|\mathrm{nhd}(G)|}, & G \in \mathrm{nhd}, \\ 0, & \text{otherwise.} \end{cases}$$

### 5: Determine summary posterior distribution

From a sample of size $S$ from the posterior distribution $P(G|D)$, distributional estimates for
structural uncertainties can be quantified by taking $\Delta$ to be an indicator function for the presence of
a given edge, given the observed data, obtained by averaging over the sample where $G^{(s)}$ is the
$s^{th}$ structure sample.

$$\Pr(\Delta | D) = \sum_{G \in \mathcal{G}} \Pr(\Delta | G, D) P(G | D)$$

$$\approx \frac{1}{S} \sum_{s=1}^{S} \Pr(\Delta | G^{(s)}, D)$$

quantifies the posterior probability $\hat{\pi}$ for the presence of an edge conditional
on the observerational data

**Figure 2.** Schematic outline of structure learning strategy.





$\beta_0^i$, $\beta_{(k_j, \tau_j)}^i$), and $\tilde{\tau}_i^{-2}$, with hyperparameters $a_\tau = 1.5$, $b_\tau = 20$, and $\nu_i^2 = 3$ (panel 2). Estimation of the graph structure and parameters proceeds in two stages, by writing the joint posterior distribution as a product of conditional distributions (panel 3). The prior specification yields an analytically tractable marginal likelihood (panel 4), thus allowing the MC$^3$ scheme of Madigan

et al. (1995) to be used to sample from the posterior distribution over possible graphs $P(G|D)$ (panel 5). Structurally modular, uniform priors over possible parent sets with lags of at most $\tau_{\max} = 6$ months and size at most $p_{\max} = 10$ are used (panel 6). A uniform proposal distribution over the neighborhood of the current graph is used (panel 7), where the neighborhood of a given graph consists of those graphs that can be obtained from it via addition or deletion of a single edge or exchange of two edges (Grzegorczyk and Husmeier, 2011), subject to the constraints on the maximum lag and parent set size. Posterior samples

for each index were obtained by running 8 chains of length $1 \times 10^7$ samples, discarding the first 250,000 samples as a burn-in period, to reduce possible biases associated with samples drawn well before the chain has converged to the target distribution. Chain convergence was assessed by considering the homogeneity of the distribution of parent sets within chains using $\chi^2$ and Kolmogorov-Smirnov tests (Brooks et al., 2003) for each index, in addition to trace plots for individual edge indicators. Results presented here are based on the full set of posterior samples.

From the sample of graph structures, estimates for the posterior distribution of quantities of interest $\Delta$ may be obtained by averaging over the sample (Draper, 1995) (panel 9 Figure 2). In particular, structural uncertainties may be quantified via an indicator function $\Delta$ for the presence of a given edge, with $\Pr(\Delta|D)$ quantifying the posterior probability $\hat{\pi}$ for the presence of that edge, conditional on the observed data $D$. In the following we summarize edges for which $\hat{\pi} > 0.5$ (both overall and within IPO phases) as summary graphs. It should be noted, however, that these figures summarize the posterior distribution

over the candidate VAR models, and do not correspond to any single model, such as the maximum a posteriori (MAP) model.

## 4 Results and discussion

Before exploring the impact of the IPO phase on inferred dependence relationships, we first assessed the tropical-extratropical causal networks in ERA5 as compared to those in similar state of the art reanalyses as reported previously (Harries and O'Kane, 2021; O'Kane et al., 2024). When the full ERA5 time period is used and the IPO excluded, similar networks were found to

those reported for JRA55 and NNR1 (O'Kane et al., 2024) (Figure 1b, compare their Figure 2), confirming that ERA5 is consistent with these previous results. Relationships associated with the MJO and ENSO are generally similar, while there are minor differences (mostly at longer lags) with respect to relationships involved lagged extratropical modes. Such differences can arise in part due to various climate model configurations, data assimilation schemes and or observing system inputs.

Inclusion of the IPO index modifies the posterior edge probabilities identified with tropical interactions (Figure 1c), most

notably enhancing the long autocorrelation time-scales associated with ENSO and MJO indices relative to those of Figure 1b. There is substantial posterior weight ($\hat{\pi} > 0.7$) assigned to dependence of the IPO on $IPO_{t-6}$ however, the greatest weight is found at one month lag with linkage to ENSO and coupling at $t-1$ and $t-2$ to both MJO indices as well as the IOD (Figure 1b). In particular, Figure 1c shows the strong tropical coupling of the IPO via central to eastern Pacific SST variations evident in nodes $IPO_{t-1} \rightarrow MEI_{t=0}$ and to the maritime continent via $IPO_{t-2} \rightarrow RMM2_{t=0}$. The overall set of relationships with all



indices is quite similar whether the IPO is included or not, an indication of the level of strong interdependence of the IPO to the tropical drivers, especially considering when extratropical activity associated with the IPO is high.

     The graphs in Figure 1 indicate Indian and Pacific oceans linkages occur via the MJO (RMM2). Specifically in Figure 1b), we see the $IOD_{t-5}$ connecting to $RMM2_t$ child and the $RMM1_{t-6}$ and $RMM2_{t-2}$ to the $IOD_t$ child. The MEI has multiple linkages to $RMM1_t$ with $RMM1_{t-4}$ with $\hat{\pi} \geq 0.9$. The MEI also links to the $IOD_{t-1}$ ($0.5 < \hat{\pi} < 0.6$) and $RMM2_{t-2}$

($0.8 < \hat{\pi} < 0.9$). These relationships are indicative of the Walker circulation and convection over the maritime continent on intraseasonal to interannual timescales as the major mechanism teleconnecting tropical sea surface temperatures in the Pacific and Indian oceans. Addition of the IPO (Figure 1c) enhances the MJO autocorrelation while also adding an edge from $IPO_{t-1}$ to the $MEI_t$ child node ($\hat{\pi} \geq 0.9$) but otherwise the graph is unchanged. The large edge weight between $IPO_{t-1}$ and $MEI_t$ is not surprising given the definition of the IPO index Eq.1.

Removing the MJO child nodes (Figure 1d) eliminates a crucial branch of the Walker circulation but does not lead to posterior edges not present in the more complete graphs. The directionality of the high probability $NAO-_{t-1}$ to $RMM2_t$ reflects the lagged influence of extra-tropical NH variability on the tropics (Kitsios et al., 2019; Stan et al., 2017). Given there are no NH child nodes present, the influence of the tropics on the NAO, i.e., a directed edge from the tropical modes to the NAO child, is not captured. In general we see that, apart from the high latitude $AO$, the mid-latitude NH extra-tropical modes (NAO-, PNA,

SCAND) are still present however in the absence of the MJO, their tropical teleconnection transfers to the MEI. The SH PSA and SAM modes are largely unaffected as they teleconnect to the tropics through ENSO directly. Posterior edges between the hemispheres (sans IPO) were comprehensively examined in the study of O'Kane et al. (2024).

     The MJO is known to couple to the extratropics via Rossby wave activity, and in particular to the NAO (Kitsios et al., 2019; Stan et al., 2017). From the result shown in Figure 1d, we observe no IPO autocorrelation beyond the $t-1$ parent node, rather

there is a $> 0.9$ edge weight from the $MEI_{t-1, \& t-6}$ months. This is in contrast to Figure 1b & c, where the MJO teleconnection was observed to mediate the tropical-extratropical interactions. In the absence of the MJO, the $MEI_{t-1, \& t-6}$ nodes (and to a lesser degree the IOD node at lag one) links with high probability to the IPO at lag one. Comparison of Figure 1c & d and the relationship between the IPO and the MEI (and IOD), when MJO variability is or is not included, reveals the important role of the MJO in linking the tropics with the extratropics, not only on the intra-seasonal time scale that dominates the regions of the

maritime continent and tropical Indian Ocean, but as being key to inter-annual ENSO and longer timescale IPO phases.

     Comparison of ERA5 networks conditioned on specified phases of the IPO, including or excluding the MJO, are shown in Figure 3 and Figure 4 for the negative (-IPO: Figure 3a and Figure 4a) and positive (+IPO: Figure 3c and Figure 4b) phases, respectively. Where the MJO is not included, it is striking the reduction and/or reallocation that occurs between tropical child indices and their parent nodes, and even more dramatic relative to that which occurred in examining all years (Figure 1).

In the -IPO case when MJO indices are not included (Figure 4a), almost no parent-child relationships remained apart from autocorrelation in the principal tropical modes i.e., IOD, MEI and IPO; the only NH extratropical parent node occurs for the $PNA_{t-4}$. The DBNs for the +IPO phase depict a similar collapse of the IPO autocorrelation with no dependence on any extratropical modes. When the MJO is included in the -IPO network (Figure 3a), there are generally more interactions at longer time lags occurring for the $PNA_{t-6}$, $SAM_{t-5}$ and $IPO_{t-6,t-5}$. Also, during the +IPO phase (Figure 3c) a short term



$t-1$ lagged relationship is noted between the NAO$^-$ and MJO with a probability between 0.7 and 0.8. The IOD in the +IPO phase is seen to have more parents and stronger evidence for the individual linkages, in particular between the MJO and the IOD and ENSO. This may be expected during the positive IPO phase when higher than average tropical SSTs in regions of the Indian/Pacific oceans are known to induce enhanced tropical-extratropical teleconnections and more frequent and larger amplitude El Niño events. Heidemann et al. (2023) found eastern Pacific El Nino's were observed (lasted) 23% (10 months) of the time during +IPO periods compared to only 14% (7 months) during -IPO periods. Furthermore, when IPO and ENSO are in phase, teleconnections around the Pacific basin are strongly enhanced (Yeh et al., 2018).

Differences between the IPO phase dependent ERA5 networks inferred during the boreal winter (December-January-Feburary: DJF) (Figure 3b & d) to those estimated over all months (Figures 3a & c) most notably occur during the +IPO phase (Figures 3c & d), and in particular a weakened lagged influence of the IOD$_{t-4,...,t-6}$ on the MJO (RMM2$_t$). Additionally we see weaker autocorrelation in the tropical modes with evidence of influence from synoptic scale blocking activity (SCAND$_{t-5}$) and zonal tropospheric flow (AO$_{t-5}$) in the NH at longer lag, presumably as a consequence of the seasonally dependent background state. In contrast, there are less substantial differences between the -IPO phase dependent ERA5 networks conditioned over all months (Figure 3a) and those during the boreal winter (Figure 3b) where the flow is more strongly influenced by the SAM$_{t-1}$ at the expense of the PSA$_{t-5}$.

When excluding the MJO child nodes, the lack of strong autocorrelation at lags beyond IPO$_{t-1}$ (Figure 1d) can be understood as follows: the IPO index is defined such that, in the absence of extra-tropical SST influences, it will be dominated by the tropical Pacific SSTs and hence ENSO. One would therefore expect a strong autocorrelation in the MEI at most times with or without the MJO, with the IPO autocorrelation dependent on the strength of the extra-tropical SST anomalies, as our results show. Taking IPO phase and seasonality into account (Figure 3) we see that long autocorrelation in the IPO occurs for the negative phase but not the positive phase and that this is consistent for conditioning only on the boreal winter. The dominance of tropical Pacific SST variability during the +IPO phase is not unexpected as this period was one where ENSO has been observed to be most active. Removal of the MJO (Figure 4a & b) further collapses the graph structure to be dominated by the IPO, MEI & IOD.

The graphs displayed in Figure 3 summarize the relationships that, individually, have a relatively high posterior probability when considering all allowed models. Inspection of individual models allows the co-occurrence of particular edges to be examined. The structures of the MAP model for each IPO phase and season, together with the corresponding regression coefficients, are shown in Tables 4 and 6. In addition to edges with $\hat{\pi} > 0.5$, the MAP models also include relationships that result in improved fit but have weaker evidence to support their inclusion. For example, in addition to the associations involving Scandinavian blocking and the AO and the tropical modes, additional interactions for which $\hat{\pi} < 0.5$ between other extratropical modes and tropical modes are found in the MAP model for the boreal winter during the positive IPO phase.

Lastly, we turn our attention to the results for the identified IPO regimes in the ACCESSCM2 climate model. As noted in Section 2, daily data is not available from which to derive MJO indices and so are not included in summary statistics for comparison to the reanalysis. Instead, our aim is to examine the potential diversity of networks across IPO phases for a number of cases. This is not possible using reanalyses due to the restricted time period over which observations are available.





**Figure 3.** (a) Posterior probabilities for edges from ERA5 data over the (a) negative IPO phase 1948-1976 and (c) the positive IPO phase considering all months; (b) & (d) considering only the boreal winter (DJF).



While our focus here is to examine changes to the summary statistics during IPO phase transitions within a modeling context, we also compare to those we have already identified in the ERA5 reanalysis to ascertain potential model biases including autocorrelation and the range of displayed internal variability.

Like many models, the ACCESSCM2 model exhibits bias in the SST spatial pattern representing the leading principal component from singular value decomposition analysis; specifically an ENSO-like pattern that extends too far west and somewhat
equatorially confined relative to ERA5. There are in addition regional SST biases, for example, the somewhat weak SSTA anomalies over the Tropical Indian Ocean also observed in other CMIP6 models (Chinta et al., 2023). That said, the availability of an ensemble of ACCESSCM2 historical simulations enables qualitative investigation of the potential diversity of teleconnections over recent decades. In particular, we are interested in the ENSO autocorrelation relative to the IPO autocorrelation during +IPO and -IPO phases, and in comparison to the case where the IPO is not included in the DBN model. Despite
being unable to include an MJO index, we take the opportunity to compare CGCM DBNs to ERA5 DBNs with and without the MJO so as to infer the role of the MJO in communicating extratropical disturbances to organized tropical intraseasonal variability, e.g., Kitsios et al. (2019) and discussions therein.

Described in Table 2 are the three ACCESSCM2 cases of IPO transition considered, namely negative directly to positive (ACCESSCM2R1: Figure 4 c & d), positive directly to negative (ACCESSCM2R5: Figure 5 e & f), and a 19 year neutral
period prior to transitioning from negative to positive (ACCESSCM2R7: Figure 5 g & h). These differences directly impact the graphs as the climatic features that "trigger" the transitions would be expected to be quite different. It is apparent that there is considerable diversity in the DBNs across each of the transition cases considered. The extent and weight of the simulated tropical-extratropical posterior edge probabilities is greater than those seen in ERA5 (MJO excluded). In particular, the extratropical indices such as the PNA, NAO$^{+,-}$ and AR parent nodes at long lags (3-6 months) with high posterior edge probability
to tropical indices (MEI, IPO) are particularly evident in the -IPO phase relative to the +IPO phase. These differences are especially apparent in the ACCESSCM2R5 and ACCESSCM2R7 IPO transitions displaying somewhat similar posterior edges e.g., to the PNA, NAO$^{+,-}$. In the ACCESSCM2R5 model -IPO phase we observe a high posterior edge probability between the IPO$_{t-1}$ and the IOD not seen in the other five model based DAGs. During the -IPO, associated with relatively cool SSTs in the equatorial Pacific, the SSTs of the eastern Indian ocean are in phase with the north-west Pacific and further associated
with relatively high temporal standard-deviations and a plausible inter-basin teleconnection across the sea of islands and the maritime continent. However, if the MJO were to be included in the ACCESSCM2 model graphs, then we would expect there to be evidence of the MJO interacting with the other modes, as was found with the ERA5 case albeit in both IPO phases (RMM2$_{t-1}$ and RMM1$_{t-6}$ -/+ IPO, respectively). The observed ERA5 IOD$_{t-1,t-2}$ parent nodes link to the MEI$_{t=0}$ child, present in the -IPO phase but not in the +IPO, does not appear in any of the modeled networks.

**5 Conclusions**

Our objective in this study has been to better characterize regime dependencies via Granger causal graphs corresponding to observed and simulated phases of multi-decadal variations in the IPO. We find not only lagged autoregressive effects but also





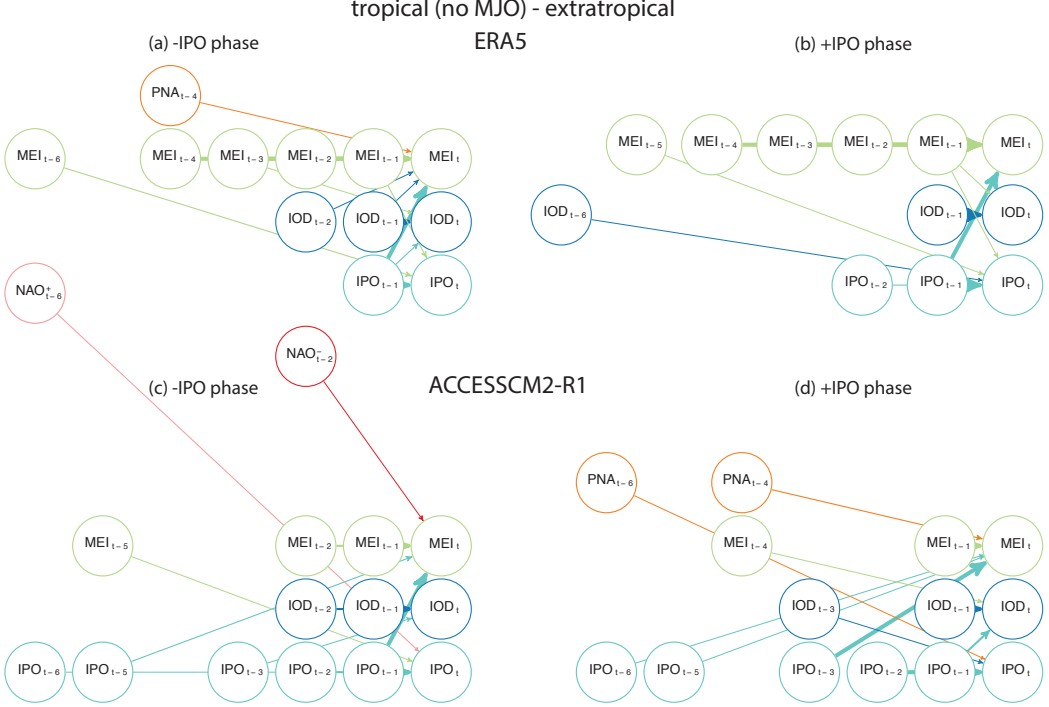

**Figure 4.** Posterior probabilities for edges from reanalysis and ACCESSCM2 GCM data over historical simulations corresponding to negative (a) ERA5 (c) R1, and positive (b) ERA5, (d) R1; IPO phases described in Table 2.

changes in tropical-extratropical relationships due to dependencies on the phase of the large scale background state, including high confidence levels for those estimates.

Our analysis noted a degree of consistency between ERA5 and earlier published results for NNR1 and JRA55 when conditioned over the same period and tropical regions. Not only does this provide confidence in the results, but it underscores the important role of data assimilation in reanalyses required to constrain the phase relationships of the internal modes of variability. When conditioned on the 1948-1998 time series, inclusion of the IPO index into the considered set of input random variables does not significantly change the posterior edge weights, whereas removal of the MJO impacts both the ENSO and

IPO autocorrelation with additional loss of causal relationships between the AO and modification to the PNA teleconnection to the tropics. The latter case reveals the importance of the MJO as a conduit to communicate extratropical variability to the tropics.

    Substantive systematic differences between the graphs for positive and negative phases of the ERA5 IPO are found to occur, most notably in the IOD and IPO autocorrelation. IPO autocorrelation for the -IPO phase extends out to $t-6$ with weaker

evidence of interactions between the MJO and IOD. For the +IPO phase, IPO autocorrelation reduces to $t-2$ but the IOD autocorrelation is very much enhanced with substantial posterior edge weights observed between the MJO and IOD at longer



**Figure 5.** Posterior probabilities for edges from ACCESSCM2 GCM data over historical simulations corresponding to negative (e) R5, (g) R7 and positive (f) R5, (h) R7; IPO phases described in Table 2.





lag. Where the MJO is removed from the input data, the DBN structures collapse to largely IOD-ENSO-IPO nodes further indicating the importance of tropical convection over the maritime continent.

Summarising the analysis of DAGs conditioned on ERA5:

– Inclusion of the IPO index does not substantially modify networks when conditioned over -/+IPO phases

    – Networks conditioned on the -IPO phase are largely similar to those conditioned on both phases

    – Networks conditioned on the +IPO phase are substantially different to the -IPO, showing increased IOD autocorrelations; reduced IPO autocorrelation; enhanced lagged posterior edge weights from MJO parent nodes to the IOD and ENSO

    – Exclusion of MJO indices removes important tropical-extratropical linkages while increasing posterior edge weights
340       between ENSO parent nodes and the IPO and IOD child nodes

    – Networks conditioned on the boreal winter are sparse with respect to those conditioned on all months of the year

Finally, we attempt to estimate regime dependencies under several transition scenarios. Due to the restricted observational record, we employ the ACCESSCM2 ocean-atmosphere coupled model ensemble simulations with historical radiative forcing. Here we do not expect the model simulations to reproduce the observed phases of the internal modes of variability, however
we were surprised by the diversity of IPO phase transitions given the same external forcing applied. A further restriction was the unavailability of full historical period daily data with which to calculate an MJO index, the omission of long continuous set of daily data critical for many studies is a common issue for CMIP submissions. In general, we found DBNs for the three considered cases to be quite diverse with teleconnections to the large scale synoptic tropospheric modes of the Northern Hemisphere i.e., PNA, NAO & AO; and long IPO autocorrelation but systematically much weaker ENSO autocorrelation, that
are not reflected in the ERA5 -IPO→+IPO transition. This in part reflects known biases in ACCESSCM2 ENSO power density spectrum (Bi et al., 2020), but also indicates the need to develop systematic causal inference methodologies to better understand regime dependencies manifest in the coupling and phase relationships between the major modes of internal climate variability.

*Code and data availability.* The hourly ERA5 pressure level data was obtained from Hersbach et al. (2023a) and single level data Hersbach et al. (2023b) using the Climate Data Store (CDS) Application Program Interface (API) described at https://cds.climate.copernicus.eu/how-to-
api. The NOAA ERSST V5 (Henley et al., 2015) based IPOTPI unfiltered data was obtained from https://psl.noaa.gov/data/timeseries/IPOTPI/. The hourly and monthly ACCESSCM2 (Bi et al., 2020) data used in this study are available from https://pcmdi.llnl.gov/CMIP6/, however, we were able to directly access the raw data from our pre-existing archive as the model and the data it generates were available locally. Harries (2020) and Collier (2024) provide all source code used to perform the analyses and results presented in this study. Data and scripts are temporarily available via the following link during peer-review and will be fully archived at Zenodo at acceptance:https://zenodo.org/records/
13999342?preview=1&token=eyJhbGciOiJIUzUxMiJ9.eyJpZCI6ImVlOTI1ZTg2LWJiY2ItNDU0My05YTcwLTdmZDgxZDM1NDYz
   NyIsImRhdGEiOnt9LCJyYW5kb20iOiI1NWI1MTc5NTdkM2IwYTQ1MmUzNGM4YzIwNzlkNjZFfXROkTwUShzZgAgrxXokTbPD
   3cuGffeEF1GzRlGiOodFJr1aSuB3zGMOl2MSQclpxoLe8OAokQNsgvWDKBqIA





*Author contributions.* MAC performed the calculations, MAC and DH produced the code on Zenodo, MAC & TJO wrote the first draft, all authors contributed to investigations, revisions and final manuscript.

*Competing interests.* None

*Acknowledgements.* The authors thank CSIRO High Performance Computing and NCI computing facility (nci.org.au) for computational and storage support. This work was supported by the Australian Commonwealth Scientific and Industrial Organisation (CSIRO) Modelling the Earth System group.





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



**Table 3.** ERA5 MAP parent sets for the IPO and tropical climate indices across all months (ALL), and for the boreal winter (DJF) conditioned on the -IPO phase for fits with $a_\tau = 1.5$, $b_\tau = 20$, and $\nu^2 = 3$, showing the estimated posterior mean $\hat{\beta}$ conditional on the MAP structure, and the 95% posterior HDI. Dashes indicate a node that is not in the MAP parent set for a given phase and or season.

| | -IPO ALL | | -IPO DJF | |
|---|---|---|---|---|
| Child node | | | | |
| Parent node | $\hat{\pi}$ | $\hat{\beta}$ (95% HDI) | $\hat{\pi}$ | $\hat{\beta}$ (95% HDI) |
| $IOD_t$ | | | | |
| $AO_{t-3}$ | 0.305 | $0.085\,(0.026, 0.144)$ | 0.091 | - |
| $AO_{t-5}$ | 0.032 | - | 0.294 | $0.210\,(0.035, 0.385)$ |
| $IOD_{t-1}$ | 1.000 | $0.620\,(0.545, 0.694)$ | 0.997 | $0.501\,(0.398, 0.607)$ |
| $IPO_{t-3}$ | 0.087 | - | 0.388 | $0.412\,(0.300, 0.526)$ |
| $MEI_{t-3}$ | 0.592 | $0.278\,(0.203, 0.354)$ | 0.136 | - |
| NHTELE3/NAO-$_{t-3}$ | 0.023 | - | 0.196 | $0.140\,(0.047, 0.238)$ |
| NHTELE4/SCAND$_{t-4}$ | 0.107 | - | 0.166 | $0.103\,(0.036, 0.170)$ |
| $PSA2_{t-3}$ | 0.029 | - | 0.198 | $0.144\,(0.064, 0.222)$ |
| $RMM1_{t-5}$ | 0.115 | - | 0.264 | $-0.123\,(-0.206, -0.041)$ |
| $RMM1_{t-6}$ | 0.030 | - | 0.825 | $-0.155\,(-0.233, -0.075)$ |
| $RMM2_{t-1}$ | 0.999 | $0.160\,(0.103, 0.218)$ | 0.024 | - |
| $RMM2_{t-2}$ | 0.474 | $0.098\,(0.041, 0.157)$ | 0.249 | $0.149\,(0.071, 0.232)$ |
| $RMM2_{t-3}$ | 0.764 | $0.117\,(0.060, 0.177)$ | 0.039 | - |
| $SAM_{t-6}$ | 0.050 | - | 0.756 | $0.132\,(0.058, 0.207)$ |
| $IPO_t$ | | | | |
| $IOD_{t-4}$ | 0.050 | - | 0.452 | $-0.119\,(-0.177, -0.058)$ |
| $IPO_{t-1}$ | 1.000 | $0.927\,(0.870, 0.984)$ | 1.000 | $0.629\,(0.441, 0.815)$ |
| $IPO_{t-2}$ | 0.392 | - | 0.250 | $0.189\,(-0.010, 0.39)$ |
| $IPO_{t-6}$ | 0.728 | $-0.142\,(-0.195, -0.092)$ | 0.971 | $-0.286\,(-0.403, -0.173)$ |
| $MEI_{t-2}$ | 0.181 | - | 0.961 | $0.368\,(0.230, 0.517)$ |
| $MEI_{t-3}$ | 0.389 | $0.126\,(0.056, 0.199)$ | 0.119 | - |
| $PNA_{t-2}$ | 0.040 | - | 0.201 | $0.063\,(0.013, 0.111)$ |
| $RMM2_{t-1}$ | 0.923 | $0.073\,(0.039, 0.104)$ | 0.079 | - |
| $RMM2_{t-2}$ | 0.472 | $0.059\,(0.024, 0.093)$ | 0.086 | - |
| $RMM2_{t-5}$ | 0.027 | - | 0.585 | $-0.148\,(-0.229, -0.063)$ |
| $MEI_t$ | | | | |
| $IOD_{t-1}$ | 0.484 | $0.085\,(0.036, 0.134)$ | 0.065 | - |
| $IOD_{t-2}$ | 0.632 | $-0.108\,(-0.156, -0.060)$ | 0.539 | $-0.118\,(-0.169, -0.068)$ |
| $IPO_{t-1}$ | 1.000 | $0.290\,(0.215, 0.360)$ | 0.414 | - |
| $IPO_{t-3}$ | 0.030 | - | 0.585 | $0.138\,(0.054, 0.221)$ |
| $MEI_{t-1}$ | 1.000 | $1.066\,(0.965, 1.171)$ | 1.000 | $0.905\,(0.802, 1.005)$ |
| $MEI_{t-2}$ | 1.000 | $-0.611\,(-0.750, -0.469)$ | 0.130 | - |
| $MEI_{t-3}$ | 1.000 | $0.421\,(0.279, 0.558)$ | 0.094 | - |
| $MEI_{t-4}$ | 0.954 | $-0.231\,(-0.320, -0.141)$ | 0.148 | - |
| $MEI_{t-5}$ | 0.080 | - | 0.200 | $0.104\,(-0.001, 0.209)$ |
| $MEI_{t-6}$ | 0.152 | - | 0.596 | $-0.167\,(-0.249, -0.081)$ |
| NHTELE3/NAO-$_{t-4}$ | 0.162 | $-0.043\,(-0.072, -0.013)$ | 0.024 | - |
| $RMM2_{t-3}$ | 0.653 | $0.061\,(0.031, 0.092)$ | 0.043 | - |
| $SAM_{t-6}$ | 0.516 | $-0.054\,(-0.083, -0.024)$ | 0.057 | - |
| $PNA_{t-3}$ | 0.012 | - | 0.183 | $0.079\,(0.028, 0.131)$ |
| $PSA2_{t-2}$ | 0.016 | - | 0.271 | $-0.067\,(-0.104, -0.027)$ |
| $PSA2_{t-5}$ | 0.071 | - | 0.475 | $0.075\,(0.035, 0.113)$ |





**Table 4.** Table 3 cont.

| | -IPO ALL | | -IPO DJF | |
|---|---|---|---|---|
| Child node | | | | |
| Parent node | $\hat{\pi}$ | $\hat{\beta}$ (95% HDI) | $\hat{\pi}$ | $\hat{\beta}$ (95% HDI) |
| $\text{RMM1}_t$ | | | | |
| $\text{AO}_{t-3}$ | 0.018 | - | 0.262 | $0.403\,(0.142, 0.65)$ |
| $\text{AO}_{t-6}$ | 0.453 | $0.198\,(0.081, 0.315)$ | 0.169 | - |
| $\text{IPO}_{t-5}$ | 0.622 | $0.390\,(0.163, 0.628)$ | 0.119 | - |
| $\text{MEI}_{t-4}$ | 1.000 | $-0.807\,(-1.009, -0.606)$ | 0.308 | - |
| $\text{MEI}_{t-6}$ | 0.838 | $0.364\,(0.180, 0.553)$ | 0.130 | - |
| $\text{PNA}_{t-3}$ | 0.018 | - | 0.162 | $0.311\,(0.071, 0.538)$ |
| $\text{PNA}_{t-6}$ | 0.752 | $0.264\,(0.146, 0.383)$ | 0.107 | - |
| $\text{RMM1}_{t-3}$ | 0.989 | $-0.206\,(-0.294, -0.116)$ | 0.142 | - |
| $\text{RMM1}_{t-4}$ | 1.000 | $-0.252\,(-0.342, -0.163)$ | 0.993 | $-0.319\,(-0.478, -0.165)$ |
| $\text{RMM2}_{t-2}$ | 0.021 | - | 0.305 | $-0.265\,(-0.420, -0.104)$ |
| $\text{RMM2}_t$ | | | | |
| $\text{IOD}_{t-2}$ | 0.702 | $-0.269\,(-0.368, -0.175)$ | 0.908 | $-0.765\,(-1.038, -0.496)$ |
| $\text{IPO}_{t-3}$ | 0.097 | - | 0.153 | $0.820\,(0.111, 1.523)$ |
| $\text{IPO}_{t-4}$ | 0.088 | - | 0.135 | $-1.075\,(-1.765, -0.419)$ |
| $\text{MEI}_{t-2}$ | 0.262 | - | 0.453 | $0.557\,(-0.002, 1.111)$ |
| NHTELE3/NAO-$_{t-3}$ | 0.036 | - | 0.343 | $0.235\,(-0.005, 0.480)$ |
| $\text{PNA}_{t-6}$ | 0.029 | - | 0.378 | $-0.649\,(-1.267, -0.063)$ |
| $\text{PSA2}_{t-1}$ | 0.032 | - | 0.258 | $-0.455\,(-0.694, -0.201)$ |
| $\text{PSA2}_{t-5}$ | 0.835 | $0.188\,(0.090, 0.283)$ | 0.044 | - |
| $\text{RMM1}_{t-1}$ | 0.642 | $0.173\,(0.075, 0.267)$ | 0.043 | - |
| $\text{RMM1}_{t-4}$ | 0.038 | - | 0.160 | $-0.341\,(-0.556, -0.125)$ |
| $\text{RMM2}_{t-1}$ | 0.168 | - | 0.636 | $-0.201\,(-0.390, -0.022)$ |
| $\text{SAM}_{t-1}$ | 0.034 | - | 0.631 | $-0.599\,(-0.881, -0.315)$ |



**Table 5.** ERA5 MAP parent sets for the IPO and tropical climate indices across all months (ALL), and for the boreal winter (DJF) conditioned on the +IPO phase for fits with $a_\tau = 1.5$, $b_\tau = 20$, and $\nu^2 = 3$, showing the estimated posterior mean $\hat{\beta}$ conditional on the MAP structure, and the 95% posterior HDI. Note. Dashes indicate a node that is not in the MAP parent set for a given phase and or season.

| | +IPO ALL | | +IPO DJF | |
|---|---|---|---|---|
| Child node | | | | |
| Parent node | $\hat{\pi}$ | $\hat{\beta}$ (95% HDI) | $\hat{\pi}$ | $\hat{\beta}$ (95% HDI) |
| $IOD_t$ | | | | |
| $AO_{t-1}$ | 0.045 | - | 0.268 | $0.268\,(0.151, 0.383)$ |
| $AO_{t-4}$ | 0.047 | - | 0.196 | $-0.186\,(-0.319, -0.055)$ |
| $IOD_{t-1}$ | 1.000 | $0.792\,(0.723, 0.859)$ | 1.000 | $0.662\,(0.564, 0.764)$ |
| $IPO_{t-1}$ | 0.476 | $0.155\,(0.07, 0.235)$ | 0.110 | - |
| $MEI_{t-5}$ | 0.324 | $-0.122\,(-0.196, -0.041)$ | 0.152 | $0.219\,(0.101, 0.341)$ |
| $NHTELE1/AR_{t-1}$ | 0.032 | - | 0.397 | $0.330\,(0.196, 0.462)$ |
| $NHTELE3/NAO\text{-}_{t-2}$ | 0.026 | - | 0.286 | $-0.179\,(-0.269, -0.093)$ |
| $PSA1_{t-5}$ | 0.601 | $-0.095\,(-0.158, -0.037)$ | 0.068 | - |
| $RMM1_{t-1}$ | 0.073 | - | 0.366 | $0.147\,(0.066, 0.233)$ |
| $RMM1_{t-6}$ | 0.972 | $-0.130\,(-0.19, -0.07)$ | 0.975 | $-0.274\,(-0.375, -0.176)$ |
| $IPO_t$ | | | | |
| $IOD_{t-6}$ | 0.508 | $-0.059\,(-0.093, -0.022)$ | 0.072 | - |
| $IPO_{t-1}$ | 1.000 | $1.081\,(0.969, 1.198)$ | 1.000 | $0.570\,(0.464, 0.678)$ |
| $IPO_{t-2}$ | 0.510 | $-0.163\,(-0.287, -0.035)$ | 0.147 | - |
| $MEI_{t-1}$ | 0.552 | $0.120\,(0.054, 0.187)$ | 0.991 | $0.408\,(0.284, 0.537)$ |
| $MEI_{t-5}$ | 0.600 | $-0.093\,(-0.144, -0.044)$ | 0.090 | - |
| $NHTELE2/NAO+_{t-4}$ | 0.028 | - | 0.073 | $0.064\,(0.019, 0.111)$ |
| $NHTELE3/NAO\text{-}_{t-5}$ | 0.035 | - | 0.087 | $0.082\,(0.035, 0.128)$ |
| $NHTELE4/SCAND_{t-2}$ | 0.037 | - | 0.347 | $-0.121\,(-0.17, -0.075)$ |
| $NHTELE4/SCAND_{t-5}$ | 0.059 | - | 0.721 | $0.114\,(0.064, 0.159)$ |
| $PSA1_{t-6}$ | 0.046 | - | 0.148 | $-0.063\,(-0.101, -0.026)$ |
| $PSA2_{t-6}$ | 0.104 | - | 0.265 | $0.063\,(0.022, 0.105)$ |
| $SAM_{t-3}$ | 0.027 | - | 0.155 | $0.07\,(0.024, 0.116)$ |
| $MEI_t$ | | | | |
| $AO_{t-3}$ | 0.274 | $0.051\,(0.018, 0.082)$ | 0.057 | - |
| $IOD_{t-2}$ | 0.041 | - | 0.359 | $0.071\,(0.021, 0.12)$ |
| $IPO_{t-1}$ | 1.000 | $0.256\,(0.182, 0.338)$ | 0.200 | - |
| $IPO_{t-3}$ | 0.186 | $0.101\,(0.011, 0.187)$ | 0.152 | - |
| $MEI_{t-1}$ | 1.000 | $1.214\,(1.106, 1.326)$ | 1.000 | $1.095\,(1.001, 1.185)$ |
| $MEI_{t-2}$ | 1.000 | $-0.805\,(-0.971, -0.635)$ | 0.210 | - |
| $MEI_{t-3}$ | 0.972 | $0.413\,(0.244, 0.576)$ | 0.211 | - |
| $MEI_{t-4}$ | 0.874 | $-0.255\,(-0.36, -0.152)$ | 0.317 | - |
| $MEI_{t-5}$ | 0.140 | - | 0.740 | $-0.187\,(-0.279, -0.097)$ |
| $RMM2_{t-3}$ | 0.967 | $0.079\,(0.045, 0.113)$ | 0.050 | - |



**Table 6.** Table 5 cont...

| | +IPO ALL | | +IPO DJF | |
|---|---|---|---|---|
| **Child node** | | | | |
| Parent node | $\hat{\pi}$ | $\hat{\beta}$ (95% HDI) | $\hat{\pi}$ | $\hat{\beta}$ (95% HDI) |
| $RMM1_t$ | | | | |
| $AO_{t-1}$ | 0.067 | - | 0.059 | $-0.212\,(-0.335, -0.09)$ |
| $IPO_{t-1}$ | 0.451 | $-0.233\,(-0.395, -0.074)$ | 0.169 | - |
| $MEI_{t-1}$ | 0.078 | - | 0.687 | $-0.975\,(-1.300, -0.653)$ |
| $MEI_{t-4}$ | 0.711 | $-0.644\,(-0.979, -0.315)$ | 0.095 | - |
| $MEI_{t-5}$ | 0.768 | $0.749\,(0.439, 1.04)$ | 0.428 | $0.670\,(0.372, 0.971)$ |
| $NHTELE1/AR_{t-3}$ | 0.020 | - | 0.184 | $0.197\,(0.063, 0.333)$ |
| $NHTELE2/NAO+_{t-6}$ | 0.019 | - | 0.104 | $0.337\,(0.178, 0.500)$ |
| $PNA_{t-2}$ | 0.130 | - | 0.395 | $-0.315\,(-0.447, -0.181)$ |
| $PSA1_{t-3}$ | 0.090 | - | 0.123 | $-0.335\,(-0.478, -0.191)$ |
| $PSA2_{t-6}$ | 0.021 | - | 0.170 | $-0.225\,(-0.351, -0.098)$ |
| $RMM1_{t-1}$ | 0.114 | - | 0.680 | $-0.410\,(-0.560, -0.259)$ |
| $RMM1_{t-3}$ | 0.998 | $-0.279\,(-0.386, -0.173)$ | 0.056 | - |
| $RMM1_{t-4}$ | 0.995 | $-0.242\,(-0.348, -0.138)$ | 0.959 | $-0.469\,(-0.602, -0.331)$ |
| $RMM2_{t-1}$ | 0.618 | $-0.172\,(-0.288, -0.065)$ | 0.119 | - |
| $RMM2_t$ | | | | |
| $AO_{t-1}$ | 0.813 | $0.215\,(0.109, 0.319)$ | 0.143 | - |
| $AO_{t-5}$ | 0.019 | - | 0.575 | $0.428\,(0.038, 0.833)$ |
| $IOD_{t-1}$ | 0.440 | $-0.205\,(-0.331, -0.081)$ | 0.302 | $-0.872\,(-1.232, -0.503)$ |
| $IOD_{t-2}$ | 0.154 | - | 0.286 | $0.716\,(0.411, 1.034)$ |
| $IOD_{t-4}$ | 0.900 | $0.509\,(0.314, 0.709)$ | 0.049 | - |
| $IOD_{t-5}$ | 0.926 | $-0.498\,(-0.681, -0.312)$ | 0.056 | - |
| $IPO_{t-2}$ | 0.694 | $0.416\,(0.248, 0.591)$ | 0.088 | - |
| $MEI_{t-4}$ | 0.981 | $-0.656\,(-0.833, -0.479)$ | 0.117 | - |
| $NHTELE1/AR_{t-6}$ | 0.028 | - | 0.322 | $0.345\,(0.153, 0.533)$ |
| $NHTELE3/NAO\text{-}_{t-1}$ | 0.804 | $0.212\,(0.109, 0.314)$ | 0.059 | - |
| $PNA_{t-4}$ | 0.014 | - | 0.371 | $0.495\,(0.105, 0.880)$ |
| $PSA1_{t-4}$ | 0.031 | - | 0.708 | $0.234\,(0.053, 0.410)$ |
| $RMM1_{t-1}$ | 0.233 | - | 0.493 | $0.339\,(0.147, 0.522)$ |
| $RMM2_{t-1}$ | 0.847 | $-0.203\,(-0.308, -0.102)$ | 0.132 | - |
| $SAM_{t-1}$ | 0.016 | - | 0.160 | $-0.359\,(-0.580, -0.146)$ |