# Peer review of "Inferring the role of IPO phase dependencies and extratropical internal variability on the tropics"

_EGUsphere, 2025_

## Author Comment (AC1)

**"Inferring the role of Interdecadal Pacific Oscillation phase dependencies and extratropical internal variability on the tropics" ref[egusphere-2025-3948]**

Mark A. Collier, Dylan Harries, & Terence J. O'Kane October 21, 2025

**1 Overall response to reviews**

**Authors**: We thank the reviewers for their comments, time and effort. In considering and responding to their feedback we are confident that the revised manuscript is much improved.

**2 Response to reviewer #1**

Reviewer #1 The manuscript addresses causal relationships between climate mode indices. Even assuming that statistical associations between indices reflecting very distinct spatial and temporal scales of variability is a valid exercise to infer climate dynamics, in my opinion the manuscript needs to be revised for improved clarity and readability. The quality of the figures needs to be substantially improved.

**Authors**: We have made substantive efforts to improve the clarity and readability of the manuscript. Particular effort has been directed at the figures incorporating much of the reviewers suggestions.

Reviewer #1 Title: IPO should be replaced by Interdecadal Pacific Oscillation

**Authors**: Done.

Reviewer #1 The structure of section 2 is very confusing, including both data description and results before the actual description of the methodology used to produce the results. It's not clear why Figure 1a) is not a separate figure and instead is mixed with results in figures 1b)-d). Furthermore, Figure 1a) is poorly described, the legend includes text such as "mw (x5), lp, nh=7, which is not described in the caption (and it's not clear whether really needed or it's relevance). What is gained from such smoothed signals in a relatively short time series is not evident, and should be better described in the manuscript.

Authors: Figure 1 has now been split into 2 figures as recommended. New figure 1 (time-series) has been redrawn with pertinent time-series kept, labels indicating the observed phase of the IPO and annotations described in the caption. We have followed standard practice in low-pass/filtering the IPOTPI to highlight the lower frequency (decadal) variability, e.g., https://psl.noaa.gov/data/timeseries/IPOTPI/. Section 2 has been revised into subsections for improved readability.

Reviewer #1 The data description and it's presentation in table 1 needs to be improved in order to clarify how each climate index was indeed calculated and at what the temporal resolution. The minimalist caption in Table 1 should be improved. What does the \* means in RMM1 and RMM2?

Authors: Additional text and links to the codes developed to calculate the climate indices have been included. All indices are based on monthly data except for the MJO RMM indices which are calculated using daily data and the indices averaged to monthly. The \* superscript on RMM indices was used to indicate that due to the unavailability of daily ACCESS model data it was not possible to calculate the indices. The superscripts have been removed and explanatory text incorporated into the body of the manuscript.

Reviewer #1 Table 2 should be better described both in terms of the caption itself and in the text. The meaning of the "All years" column in Table 2 is not clear. More importantly, the objective criteria by which the positive and negative IPO phases are identified should be clearly and unambiguously stated.

Authors: We have added a more details in the revised caption to explain the contents of Table 2. All years" has been changed to "Available years". As described in the text, we have used a simple moving-window applied to the monthly time-series, to identify phases of the IPO of sufficient length to enable fitting of the timeseries model. We have verified that using either a moving window or retained harmonics produces very similar estimations of prolonged periods where the system is in a given IPO phase. References to ERSST in the table and text has been retained as it is a highly reliable SST dataset and was used to confirm the reliability of the ERA5 dataset in estimating IPO variability.

Reviewer #1 The posterior probability plots should be improved (Figures 1b)-d), Figures 3, 4, and 5 by adding axes (as in Figure 1b), for consistency), and particularly to improve readability of the posterior probabilities, as in the current configuration it is not possible to effectively distinguish between low and intermediate probabilities. Maybe just providing the two highest probability ranges and distinguishing between them in another way other than width of the line (for example by using solid and dashed or dotted lines) would enable to reader to actually see the results, with the current design it's almost impossible. Authors: We have added the x and y axis to all of the sub-figures as recommended. In addition we have modified the two highest probability ranges as solid lines to distinguish them from probabilities < 0.8 which are indicated by dashed lines. As any probability  $\ge 0.5$  is a reasonable indicative measure that a robust relationship exists between child and parent node, we retain these in the figures.

---

## Author Comment (AC2)

**"Inferring the role of Interdecadal Pacific Oscillation phase dependencies and extratropical internal variability on the tropics" ref[egusphere-2025-3948]**

Mark A. Collier, Dylan Harries, & Terence J. O'Kane

January 21, 2026

**1   Overall response to reviews**

**Authors**: We thank the reviewers for their comments, time and effort. In considering and responding to their feedback we are confident that the revised manuscript is much improved.

**2   Response to reviewer #2**

**Reviewer #2** This study uses Bayesian structure learning to examine causal relationships between major tropical and extratropical climate modes and identifies notable regime-dependent differences in climate dynamics. The work offers an interesting perspective on how these relationships may vary across regimes. However, in its current form, the manuscript would benefit from clearer calarify, and stronger justification of methodological choices. I therefore recommend further revision before the study can be considered for publication.

  **Authors**: We have made substantive efforts to improve the clarity and readability of the manuscript. Particular effort has been directed at the figures incorporating much of the reviewers suggestions.

  **Reviewer #2** While the Introduction provides substantial background on IPO, ENSO, and causal discovery methods, the core scientific question is not stated sharply enough. For example, it is unclear whether the primary objective is to demonstrate that causal network structures differ between IPO phases.
**Authors**: Thanks. We have revised the title and content to better clarify our objectives.

  **Reviewer #2** Several inferred links, particularly involving the MJO, SAM, and NAO, are described descriptively (e.g., as edges in the graph), but their physical interpretation remains vague. The manuscript would benefit from deeper discussion on what lagged statistical causality implies in the context of

atmospheric teleconnections and how these findings relate to established mechanisms.

**Authors**: Agreed. We have included appropriate discussion of physical interpretations in the revision.

**Reviewer #2** The classification of IPO phases is based on a 30-year moving window and manual selection (Table 2), but the specific criteria and sensitivity of this approach are not well explained. Similarly, the choice of a 6-month maximum lag in the structure learning is not justified in terms of known climate dynamics. Both choices are critical to the causal analysis and should be more clearly motivated and discussed.

**Authors**: The application of a moving window applied to the IPO tripole index is a standard approach to isolate multi-decadal variability and for identifying the respective phases of the IPO. Here we have chosen a 30-year window as we are primarily interested in the timing of the phase transition if present in the data. The most common choices of window range from 13- to 30-years. Henley et al. [2015] point out that results are qualitatively consistent across three SST reanalysis datasets (HadISST1, HadSST.3.1.0.0, ERSSTv3b) for moving windows of length varying between 10-40 years. We apply the same approach to CMIP model data to augment the number of possible alternate cases of phase transitions beyond the observed historical single case of transition between positive and negative multi-decadal regimes.

**Authors**: The choice of for networks based on monthly indices, i.e., $\tau_{max} = 6$ months, corresponds to the approximate e-folding time of the MEI and we argue this is an appropriate timescale for tropical variability. This choice also allows direct examination of the influence of IPO phase on the causal relationships between the internal modes relative to their annual and seasonal climatological DAGs recently reported by O'Kane et al. [2024] in their examination of CMIP5 models where the IPO was not absent from the parent set.

On a technical note, longer lags can necessitate requiring increased sample sizes. For each sampling method, we run multiple chains, discarding the first half of each sample as burn-in - currently 250,000 sample burn in. For the samplers considered, as sample size increases, approximate convergence of the chains to the target distribution as assessed using the $\chi^2$ and Kolmogorov-Smirnov tests may become prohibitive in terms of computation times.

**Reviewer #2** Several figures (e.g., Figures 1b–d, 3, 4, 5) are difficult to interpret due to visual overloading and unclear encoding. Posterior probabilities are represented solely by line width, which is hard to distinguish visually, especially without a legend or clear thresholding. Some axes and labels are inconsistent or missing. I suggest using line styles or colors to improve clarity.

**Authors**: Agreed. This is a fair point. Figures have been redrawn for clarity with additional line styles incorporated..

**Reviewer #2** The manuscript mixes data, methods, and results in ways that make it difficult to follow. For instance, Section 2 presents results (e.g., posterior causal networks in Figures 1b–d) before the causal inference method is properly introduced in Section 3. Figure 1a, showing IPO phase identification, also appears abruptly without a clear explanation of the smoothing/filtering

methods or its connection to the subsequent analysis. I recommend reorganizing the manuscript so that methods are presented before results, and separating Figure 1 into more coherent units with clearer captions and contextual framing. **Authors**: Some reorganization has been made and figure 1 broken up and completely revised for clarity. Additional explanation of the filtering and contextual framing has also been incorporated.

**References**

B. J. Henley, J. Gergis, D. J. Karoly, S. Power, J. Kennedy, and C. K. Folland. The Tripole Index for the Interdecadal Pacific Oscillation. *Clim. Dyn.*, 45: 3077–3090, 2015. doi:10.1007/s00382-015-2525-1. URL `https://psl.noaa.gov/data/timeseries/IPOTPI/`.

T. J. O'Kane, D. Harries, and M. A. Collier. Bayesian Structure Learning for Climate Model Evaluation. *Journal of Advances in Modeling Earth Systems*, 16:1–33, 2024. doi:10.1029/2023MS004034.